# Translucency of CAD/CAM and 3D Printable Composite Materials for Permanent Dental Restorations

**DOI:** 10.3390/polym15061443

**Published:** 2023-03-15

**Authors:** Alessandro Vichi, Dario Balestra, Nicola Scotti, Chris Louca, Gaetano Paolone

**Affiliations:** 1Dental Academy, University of Portsmouth, Portsmouth PO1 2QG, UK; 2School of Dental Medicine, Alfonso X El Sabio University, 28691 Madrid, Spain; 3Department of Surgical Sciences, Dental School Lingotto, University of Turin, 10126 Turin, Italy; 4Department of Dentistry, IRCCS San Raffaele Hospital and Dental School, Vita Salute University, 20158 Milan, Italy

**Keywords:** translucency, contrast ratio, translucency parameter, composite, cad/cam, 3d printing, resin

## Abstract

The aim of the study was to compare the translucency of CAD/CAM and printable composite materials for fixed dental prostheses (FDP). Eight A3 composite materials (7 CAD/CAM and 1 printable) for FPD were used to prepare a total of 150 specimens. CAD/CAM materials, all characterized by two different opacity levels, were: Tetric CAD (TEC) HT/MT; Shofu Block HC (SB) HT/LT; Cerasmart (CS) HT/LT; Brilliant Crios (BC) HT/LT; Grandio Bloc (GB) HT/LT; Lava Ultimate (LU) HT/LT, Katana Avencia (KAT) LT/OP. The printable system was Permanent Crown Resin. 1.0 mm-thick specimens were cut from commercial CAD/CAM blocks using a water-cooled diamond saw, or 3D printed. Measurements were performed using a benchtop spectrophotometer with an integrating sphere. Contrast Ratio (CR), Translucency Parameter (TP), and Translucency Parameter 00 (TP_00_) were calculated. One Way ANOVA followed by Tukey test for post hoc were performed for each of the translucency system. The tested materials exhibited a wide range of translucency values. CR ranged from 59 to 84, TP from 15.75 to 8.96, TP_00_ from 12.47 to 6.31. KAT(OP) and CS(HT) showed, respectively, the lowest and highest translucency for CR, TP and TP_00_. Due to the significant range of reported translucency values, clinicians should exercise caution when choosing the most appropriate material, especially considering factors such as substrate masking, and the necessary clinical thickness.

## 1. Introduction

Restorative dentistry aims to recreate natural dental structures from both functional and esthetic perspectives. The growing demand for esthetic and long-lasting dental treatments has led to the development of novel restorative solutions especially for indirect adhesive restorations. In this regard, two technologies have emerged: CAD/CAM and more recently 3D Printing (3DP).

CAD/CAM technology has continuously improved, providing a reliable and predictable process for single-unit and multiple units restorations. Today, dentists can use several materials for CAD/CAM restorations, such as glass-ceramics, zirconia, and composites [1,2]. CAD/CAM composite blocks (CCBs), also referred to as resin nano-ceramics, resin-based composites, and nanohybrid restorative blocks, offer advantages in terms of appearance, affordability, ease of fabrication, intra-oral adjustments and/or repair [3]. They also exhibit lower wear rates and greater resistance to discoloration compared to traditional resin composites, mainly due to the higher degree of conversion achieved during the manufacturing process, which is the result of a standardized industrial process where the materials are cured at high temperature and/or under pressure to maximize polymer cross-linking and improve material properties [4]. Introduced in the early 2000s, contemporary CCBs vary in terms of composition and material properties [5].

Regarding the 3DP technology, Digital Light Projection (DLP) and Stereolithography (SLA) are the most diffused manufacturing techniques. They both use light, LED for DLP and laser for SLA, to polymerize targeted areas [6] of a wide range of resin materials. These materials share a composition primarily consisting of acrylates and epoxy resin, photoinitiators and UV absorbers that enable photopolymerization [7]. 

Even if Daher et al. [6] reported that the physical characteristics of the 3D-printed dental composites belong to the category of flowable composite resin (flexural strength = 107–130 MPa; elasticity modulus = 4 GPa), further evaluation is needed to determine whether the indication for permanent single-tooth restorations, as claimed by manufacturers, is clinically valid in the medium and long term. Nevertheless, 3D-printed resin materials with the indication for permanent use have been marketed.

In order to meet the increasing aesthetic needs of patients, clinicians need to better understand the materials’ optical properties, with translucency being a crucial factor in replicating the optical behavior of enamel and dentin [7].

Translucency refers to the ability of light to pass through a material and is characterized by an intermediate state between complete opacity or transparency and has a noteworthy clinical relevance for several reasons: (i) matching dental tissues’ optical properties; (ii) masking undesired substrate colors; (iii) helping light transmission for light curing cements. Different methods can be used to calculate translucency, including Contrast Ratio (CR) and Translucency Parameter (TP, TP_00_). CR is defined as the ratio of the reflectance of a specimen placed over a black backing to that over a white one of known reflectance, and it is defined as an estimate of opacity [8]. TP is a color difference (ΔE) between a material measured over white and black backing [9]. An ISO standard for evaluating translucency is unfortunately currently not available even if some indications can be retrieved from ISO 28642:2011 [10]. The aim of this study was to compare CR, TP, and TP_00_ of CAD/CAM and printable resin composite materials for FDPs.

The tested null hypothesis was that translucency is different between distinct translucency levels (e.g., HT, LT, etc.) of the same material, but not between the same translucency of different materials.

## 2. Materials and Methods

The present study involved the selection of eight materials for FDP: seven CAD/CAM blocks for CEREC system and one 3DP material (Table 1). CAD/CAM materials’ specimens were obtained by cutting the CAD/CAM blocks with a water-cooled low-speed diamond saw (Isomet, Buehler, Lake Bluff, IL, USA). A custom-made setup was employed to maintain the alignment of the blocks with respect to the saw blade during the cutting process. Ten specimens were cut for each material (*n* = 10) with a dimension of 14 mm × 14 mm and 1.0 mm thickness (Figure 1).

For the 3D-printed material, squared specimens with a dimension of 14 mm × 14 mm and a thickness of 1.0 mm were designed using Thinkercad software (Autodesk, San Rafael, CA, USA) (Figure 2).

Then, the project was exported in stl file format and subsequently imported into PreForm software (Formlabs, Somerville, MA, USA) for automatic supports calculation and slicing (Figure 3). 

The printing parameters were set to 50-micron layer thickness using the exposure time provided by the software for the resin object of the test. Specimens were 3D printed with the Formlabs 3B 3D printer (Formlabs, Sommerville, MA, USA). After printing, specimens were removed from the printing platform, and still with raft and supports were subjected to the washing procedure for 3 min with the FormWash (Formlabs, Somerville, MA, USA), an automated washing machine, using 99% isopropyl alcohol (IPA), to remove uncured resin. After washing, the specimens were cured for 20 min at 60 °C in the FormCure (Formlabs, Somerville, MA, USA), an automatic curing machine from the same manufacturer. Then, the specimens were removed from supports, carefully sandblasted with glass bead blasting material 50 μm, (Perlablast micro, Bego, Bremen, Germany) at a pressure of 1.5 bar to remove the filler particles layered onto the surface, and then post cured again in the FormCure for another 20 min at 60°. 

Finally, all the 150 specimens were finished and polished on a grinder/polisher (minimet, Buehler, Lake Bluff, IL, USA) with wet 320-, and 600- grit silicone-carbide paper and subsequently ultrasonic-cleaned in distilled water for 10 min prior to measurement. The specimens had their thickness measured with a digital caliper (Absolute Digimatic, Mitutoyo, Tokyo, Japan). Specimens that varied more than 0.05 mm from the intended thickness (1.0 mm) were discarded.

A benchtop spectrophotometer (PSD1000, OceanOptics, Dunedin, FL, USA), equipped with an integrating sphere (ISP-REF, OceanOptics) and a 10 mm opening, was utilized in combination with the corresponding color measurement software (OOILab 1.0, OceanOptics). The D65 illuminant and 10° standard observer were employed as the measurement standards. The measurements were performed using white and black standard tiles as references.

Specimens’ translucency was calculated with contrast ratio (*CR*) and translucency parameter (TP, TP_00_):

The *L** coordinates values measured on white and black background were used to calculate the luminance from Color Space CIEXYZ, as follows:$$Y={\left(\frac{L+16}{116}\right)}^{3}\times Y_{n}$$

*Y* values of the specimens recorded on white (Yw) and black (Yb) backgrounds were used to calculate Contrast Ratio (*CR*) as follows:$$CR=\frac{\mathrm{Yb}}{\mathrm{Yw}}$$

TP was calculated using the CIEL*a*b* formula:$$\mathrm{TP}\;=\sqrt[{2}]{{\left(L{*}_{B}-L{*}_{W}\right)}^{2}+{\left(a{*}_{B}-a{*}_{W}\right)}^{2}+{\left(b{*}_{B}-b{*}_{W}\right)}^{2}}$$
where the *W* refers to CIELab values on a white background while “*B*” on black background.

TP_00_ was calculated using the CIEDE2000 formula:$${\mathrm{TP}}_{00}\;=\sqrt[{2}]{{\left(\frac{∆L^{\prime }}{K_{L}S_{L}}\right)}^{2}+{\left(\frac{∆C^{\prime }}{K_{C}S_{C}}\right)}^{2}+{\left(\frac{∆H^{\prime }}{K_{H}S_{H}}\right)}^{2}+R_{T}{\left(\frac{∆C^{\prime }}{K_{C}S_{C}}\right)}^{2}{\left(\frac{∆H^{\prime }}{K_{H}S_{H}}\right)}^{2}}$$
where Δ*L*′, Δ*C*′, and Δ*H*′ are the differences in lightness, chroma, and hue for a pair of specimens over the black and white background, respectively. *R_T_* is the rotation function that accounts for the interaction between chroma and hue differences in the blue region. *S_L_*, *S_C_*, and *S_H_* are weighting functions for adjustment of the total color difference for variation in perceived magnitude with variation in the location of the color coordinate difference between two color measurements. Parametric factors *K_L_*, *K_C_*, and *K_H_* in CIEDE2000 formula were set to 1 [11].

### Statistical Analysis

For each of the three translucency assessment calculation methods used (CR, TP, TP_00_), data were tested to fit a normal distribution with the Kolmogorov–Smirnov test and the homogeneity of variances was verified with Levene’s test. According to these preliminary tests, for each of the three translucency calculation methods a one-way ANOVA was performed, followed by the Tukey test post hoc. In all the statistical tests, the level of significance was set at *p* < 0.05. The statistical analyses were processed by SigmaPlot 11.0 (Systat Software, Inc., San Jose, CA, USA) software.

## 3. Results

Higher CR values and lower TP/TP_00_ values correspond to more opaque materials, whereas lower CR and higher TP/TP_00_ correspond to materials with higher translucency.

For CR (Table 2), the results varied from 59.0 (CS, HT) to 84.4 (KAT, OP). The lowest level of translucency was showed by KAT(OP) (84.4), statistically significantly opaquer than all the other materials. The translucency values calculated, and the statistical differences are reported in Table 2.

Figure 4 shows a graphical representation of CR values.

TP (Table 3) varied from 15.75 (CS, HT) to 8.96 (KAT, OP), while TP_00_ (Table 4) varied from 12.47 (CS, HT) and 6.31 (KAT, OP). 

Statistically significant differences were reported among the materials for TP and TP_00_. The highest level of translucency was showed by CS(HT) (TP = 15.75, TP_00_ = 12.47), statistically significantly more translucent than all the other materials. The translucency values calculated, and the statistical differences are reported in Table 3 and Table 4. 

Figure 5 and Figure 6 show a graphical representation of TP and TP_00_ values.

## 4. Discussion

Dental CAD/CAM systems have revolutionized restorative dentistry by allowing clinicians to design and fabricate high-quality restorations using computer-assisted techniques. Esthetic materials play a crucial role in the success of CAD/CAM restorations, with translucency being a key factor in achieving natural-looking results [1]. Translucency refers to the ability of a material to transmit light while diffusing it in the process. In restorative dentistry, translucency is important because it mimics the optical properties of natural teeth, which have varying degrees of translucency depending on their location and thickness. CAD/CAM systems require materials with a high degree of translucency to achieve natural-looking restorations.

One esthetic material commonly used in CAD/CAM systems is lithium disilicate glass-ceramic. This material has high translucency and can be used to fabricate crowns, veneers, and inlays. It also has good mechanical properties, such as high flexural strength, which make it suitable for use in the posterior region of the mouth [2].

Another material used in CAD/CAM systems is zirconia [2]. While zirconia is not as translucent as lithium disilicate glass-ceramic, it has excellent mechanical properties and can be used to fabricate full-contour restorations, such as crowns and bridges, in the posterior region of the mouth. Zirconia can also be layered with porcelain to achieve a more natural-looking result [2]. In recent years, hybrid materials, such as resin-ceramic composites, have also become popular in CAD/CAM systems [1,3,4]. These materials combine the esthetic properties of ceramics with the strength and durability of resin-based materials. They have good translucency and can be used to fabricate veneers, inlays, and onlays.

For all three translucency calculation methods used (CR, TP, and TP_00_), the tested materials showed a wide range of translucency values, and the differences observed were statistically significant. Therefore, the null hypothesis was accepted.

The translucency of a material plays a crucial role in choosing metal-free materials. This property is commonly evaluated through either CR or TP. CR is defined as the ratio of luminous reflectance on a black backing to the luminous reflectance on a white backing. The CR ranges from 0 to 1, with 0 indicating a completely transparent material and 1 indicating complete opacity [9]. The Translucency Parameter (TP) was introduced as a direct measure of translucency and is defined as the color difference of a material at a specified thickness when in contact with ideal black and ideal white backings [9]. TP can be calculated using both CIEL*a*b* (TP) and CIEDE2000 (TP_00_) formulas. Since translucency is dependent on thickness and there is no established ISO standard for evaluating translucency in dentistry, for this study, 1 mm thickness was used in order to compare the results with existing literature.

A huge variety of translucency values were reported for the investigated materials. In the present study, CS HT was reported to be the most translucent, followed by TEC HT. This finding is in agreement with Günal Abduljalil B. et al. [12] that reported CS to be the most translucent material when compared to GB, BC and LU. Our findings are also in agreement with Alfouzan et al. who reported decreasing TP values for A2/HT formulation of the following CCBs: CS > LU > SB.

This outcome can be tentatively explained by taking into consideration that CS is a nanoceramic material that has dispersed fillers, such as silica and barium glass, incorporated within a polymeric matrix. It does not contain any opacifying agent, the filler particles in CS are smaller than those in other materials [13], and the refractive indices of Bis-MEPP and UDMA are similar to those of silica and barium glass fillers. These factors may account for CS’s higher translucency values.

The low ΔE*ab value of CS could be due to its composition of aluminum barium silicate particles embedded in a polymer network, which enhances light transmission as no opacifying compounds are present. [14,15].

High differences in refractive index between the reinforcing filler or opacifying compounds and the polymeric matrix, result in increased opacity levels due to multiple reflections and refractions at the interface between the matrix phase [16].

The refractive indices of the Bis-GMA, UDMA, TEGDMA, Bis-EMA, TiO_2_, Al_2_O_3_, and ZrO_2_ are quoted as 1.55, 1.48, 1.53, 1.55, 1.46, 2.49, 1.77, and 2.22, respectively. Radiopaque fillers, such as those including zirconium, barium or strontium, present refractive indices of approximately 1.55 [12]. TiO_2_ has the highest refractive index, which causes a significant discrepancy with the resin matrix, thus making BC one of the opaquest materials [10]. It has been observed that Bis-GMA has a more translucent appearance compared to UDMA and TEGDMA. This is because Bis-GMA has a refractive index that is closer to silica and zirconia filler systems than UDMA and TEGDMA [17].

Haas et al. [15] analyzed the effect of various opacifiers on the transparency of experimental dental composite resins and reported the opacifying properties of metal oxides through their results all opacifiers (TiO_2_, ZrO_2_, and Al_2_O_3_ in descending order) decreased L* value. 

The increased TP values in some of the studied materials can also be attributed to the presence of silica/zirconia nanoparticles embedded in a highly cross-linked resin matrix. These nanoscale filler particles with diameters smaller than visible light wavelength result in less light scattering and improved light transmission, leading to enhanced translucency [18].

Gunal and Ulusoy [19] reported TP (CIELAB) values of 18.64, and 17.93 for a 1-mm-thick specimen of CS (LT, A2), and LU (LT, A2), respectively. 

Koenig et al. analyzed the filler fraction of various CAD/CAM RBCs (Shade A2) such as BC, CS, LU, SB, TEC and GB using different translucencies (HT, LT, MT). The authors reported fillers’ ranges from the lower nm range to a maximum size of approximately 12 µm. The largest filler particles were identified in SB, followed by LU and GB. The authors also reported negligible differences between HT and LT variants regarding filler mass and volume proportions.

Among the investigated materials, the TP values for the studied samples range between TP = 8.96 for KAT OP to TP = 15.75 for CS HT. Except for CP, all manufacturers offer two degrees of translucency, being high-translucency (HT) and low-translucency (LT) the most common formulations of the CAD/CAM blocks (LU, CS, GB, SB, BC). Exceptions are TEC which presents an HT and a medium translucency (MT) formulation, and KAT that presents an LT and a higher opacity formulation (OP) of the material. When HT formulations are considered, their values show high variability, ranging from CR = 59 for CS and CR = 66 for BC and LU, respectively, or TP = 15.75 and TP = 12.84 for BC. Interestingly the LT formulation of CS showed higher translucency (CR = 65) than the HT formulation of BC (CR = 66).

Another aspect that should be taken into clinical consideration is that translucency influences the masking capability of restorative materials. Porojan et al. [20] examined the masking effectiveness of the HT formulation of various composite CAD/CAM materials. They found that masking ability decreased in the following order: SB > LU > CS thus confirming the findings of the present study. Niu et al. [21] reported that lithium disilicate restorations with a thickness greater than 1.5 mm, the appearance is not significantly affected by the color of the base material or the cement used. Whether this could also be related to composite-based materials has yet to be defined and could be the object of further studies related to CCBs translucency. Nevertheless, when compared with other materials such as lithium disilicate or polymer infiltrated ceramic network (PICN), that is Vita Enamic (Vita Zahnfabrik, H. Rauter GmbH), CCBs show higher translucency values [22,23].

Examining the optical characteristics of dental materials and establishing a direct correlation with equivalent natural tissues can help improve the final esthetic clinical result. Translucency values for natural enamel and dentin have been reported in the literature but without conclusive evidence. The TP values of 1.0 mm thick human enamel ranged from 15 to 19 [24]. According to the result of the current study only few materials (TEC (HT), CS (HT)) can reproduce the translucency of natural enamel being their TP value of 15.3 and 15.75, respectively. Conversely, if the paper from Yu et al. is taken as a reference, none of the investigated materials can reproduce either natural enamel (TP = 18.1) or dentin’s (TP = 16.4) translucency [25]. Another study reported the mean CR for enamel and dentin to be 0.45 and 0.65, respectively [26]. If this study is taken as a reference, several investigated materials can be indicated for dentin replacement, mainly with HT formulation (SB HT, CS LT, BC HT, GB HT, LU HT). Nevertheless, although it is important to avoid excessive translucency to prevent a decrease in lightness and a grayish appearance, none of the investigated materials seem to have comparable translucency to human enamel (being all CR ≥ 0.59) [27]. 

Regarding the 3D-printed material, it shows an intermediate translucency when compared to the other CAD/CAM materials. The properties of 3DP material can be influenced by other aspects than the composition of the materials, e.g., by the printing technique [28]. However, there is currently a lack of evidence regarding the clinical performance of 3DP materials intended for permanent restorations [29]. If the composition is considered, this type of material, which is provided in a single translucency level, is characterized by a low filler content (30–50% wt.) than the other investigated materials (61–86% wt.) and by one single monomer (Bis-EMA). These characteristics are linked to the earliness of 3D-printed materials technology that still has some limitation in providing higher filler loads and monomer types. BIS-EMA monomer in fact, in respect to BisGMA, does not have hydroxyl groups, thus decreasing viscosity and making it an ideal monomer for 3D printing. Future improvements of the types and ratios between classical monomers used for conventional or CAD/CAM RBCs (such as UDMA or TEGDMA) represent a pivotal field of development for 3D printed materials for dental applications. It has been in fact reported that the accuracy of experimental 3D printing materials based on Bis-EMA can be improved with the addition of UDMA and TEGDMA [30].

As a limitation of this study, only one thickness has been investigated. Being in fact an ISO standard for evaluating translucency is still not available, only some indications can be retrieved from ISO standard 28642:2011 concerning guidance on color measurement in dentistry [10]. An increase in specimen thickness has been reported to lead to a significant reduction in translucency values and a negative exponential relationship between translucency values and restorative materials’ thickness [31,32,33,34,35,36,37,38,39]. This aspect also influences cementation as by increasing the restoration’s thickness, reduced light energy is delivered to the luting cements [40]. It has been in fact suggested that for the placement of CAD/CAM PICN restorations higher than 1.0 mm in thickness the selection of a dual-cure resin cement is a favorable option [40]. This aspect concerning light energy in relationship with thickness and translucency could be the object of future studies for CCBs and 3DP materials. Using 3DP materials for permanent restoration is an innovative field in the dental materials technology; therefore, future studies are advisable, e.g., to correlate the findings of the present study with color stability [41] of CAD/CAM and printing materials. As well, mechanical properties such as compression modulus, tensile modulus, microhardness needs to be investigated to compare 3DP materials with CAD/CAM resin composite materials [42]. Furthermore, studies involving more clinical procedures such as finishing and polishing, cementation and fatiguing shall be performed with fractographic and fractal dimension analyses, to compare CAD/CAM and printable materials, as previously reported [43].

## 5. Conclusions

Composite materials for permanent dental restorations show a wide range of translucency. Same translucency levels (HT, LT) do not have correspondence among different manufacturers. Clinicians should be cautious when selecting the appropriate material due to the wide range of reported translucency values, taking into account factors such as substrate masking and the required clinical thickness.

## Figures and Tables

**Figure 1 polymers-15-01443-f001:**
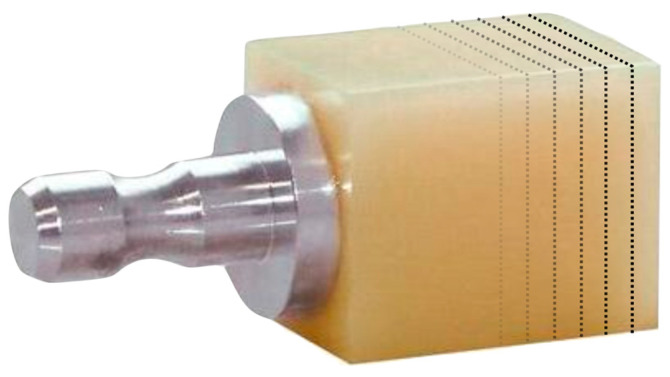
Graphical representation of the cuts performed on the CAD/CAM blocks in order to obtain specimens from different materials.

**Figure 2 polymers-15-01443-f002:**
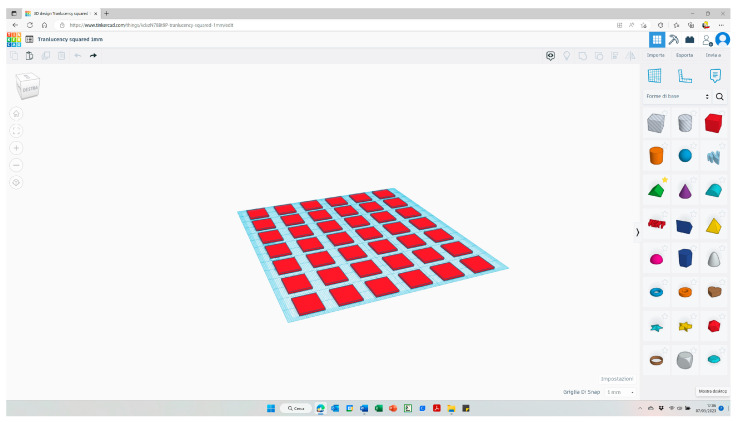
Specimens with a dimension of 14 mm × 14 mm and a thickness of 1.0 mm were designed using Thinkercad software.

**Figure 3 polymers-15-01443-f003:**
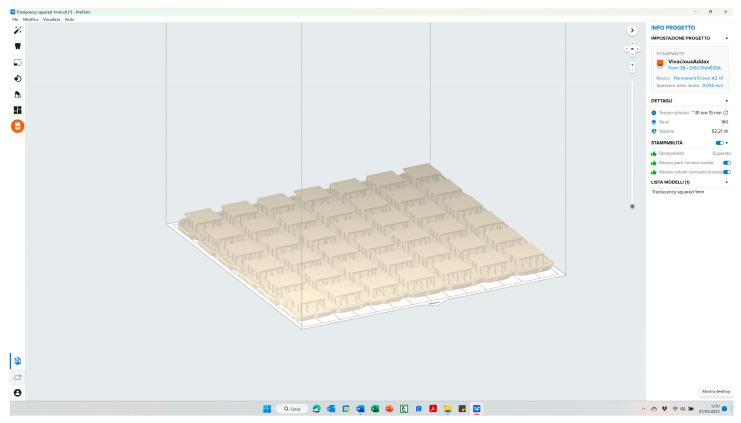
Specimens’ automatic supports calculation and slicing was performed in PreForm software.

**Figure 4 polymers-15-01443-f004:**
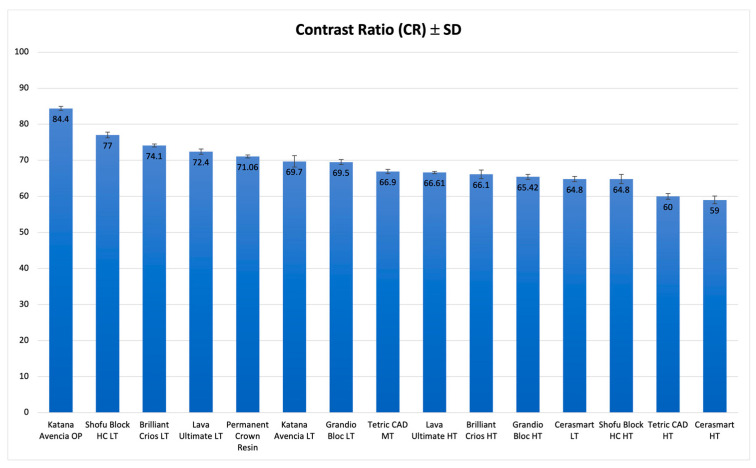
Graphic representation of CR for 1.0 mm of the investigated materials.

**Figure 5 polymers-15-01443-f005:**
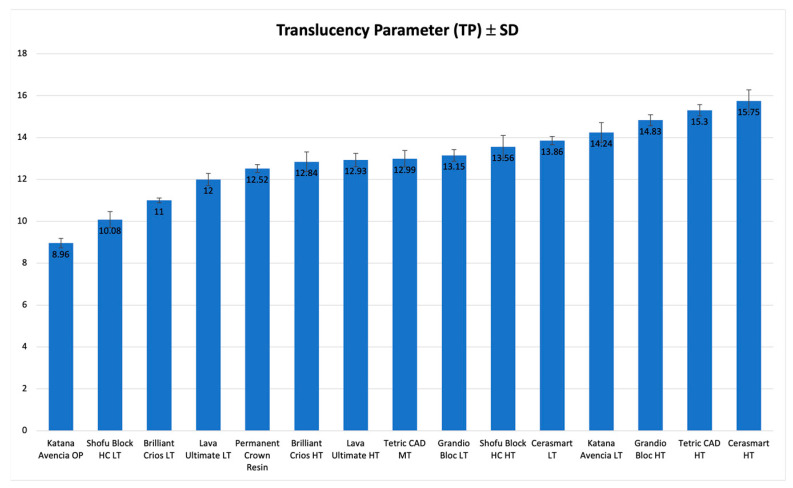
Graphic representation of TP for 1.0 mm of the investigated materials.

**Figure 6 polymers-15-01443-f006:**
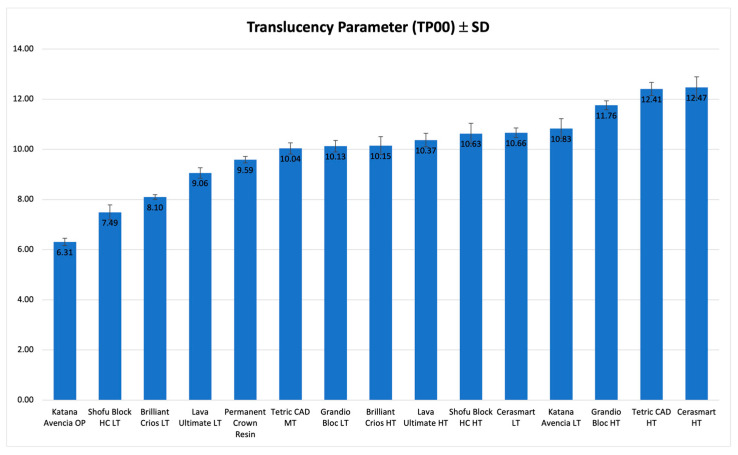
Graphic representation of TP_00_ for 1.0 mm of the investigated materials.

**Table 1 polymers-15-01443-t001:** Characteristics of the investigated materials.

Product	Type	Code	Organic Matrix	Inorganic Filler	Manufacturer	Shade	Lot.	Translucency
Lava Ultimate	CAD/CAM composite	LU	BisGMA, UDMA, BisEMA, TEGDMA (21% wt.)	zirconia-silica clusters (0.6–10 m) + zirconia (4−11 nm) + silica (<20 nm) (79% wt.)	3M ESPE, St. Paul, MN, USA	A3	N501110, N933367	LT, HT
Brilliant Crios	CAD/CAM composite	BC	Cross-linked methacrylates (Bis-GMA, Bis-EMA, TEGDMA) (30% wt.)	Inorganic part: barium glass with particle size, 1 lm and amorphous silica SiO_2_ with particle size, 20 nm (70.7% wt.)	Coltène/Whaledent, Altstatten, Switzerland	A3	60019988, 60019996	LT, HT
Cerasmart	CAD/CAM composite	CS	BisMEPP, UDMA, DMA (29%wt.)	Silica and barium glass nanoparticles (Silica (20 nm), barium glass (300 nm)) (71% wt.)	GC, Tokyo, Japan	A3	1506172, 1506171	LT, HT
Katana Avencia Block	CAD/CAM composite	KAT	UDMA, TEGDMA	aluminum oxide (20 nm), SiO_2_ (40 nm) (62% wt.)	Kuraray-Noritake, Miyoshi, Japan	A3	000302, 000584	OP, LT
Shofu HC Block	CAD/CAM composite	SB	UDMA + TEGDMA (39% wt.)	Silica-based glass and silica (61% wt.)	Shofu, Kyoto, Japan	A3	071601, 071601	LT, HT
Grandio Blocs	CAD/CAM composite	GB	UDMA + DMA (14% wt.)	Nanohybrid filler (86% wt.)	Voco, Cuxhaven, Germany	A3	1718131, 1719093	LT, HT
Tetric CAD	CAD/CAM composite	TEC	Bis-GMA, Bis-EMA, TEGDMA, UDMA	barium aluminium silicate glass with a mean particle size of <1 µm and silicon dioxide with an average particle size of <20 nm (71.1% wt.)	Ivoclar-Vivadent, Schaan, Liechtenstein	A3	X20766, Z00PD0	MT, HT
Permanent Crown Resin	Methacrylic acid ester-based resin	PCR	≥50–<75 % wt. Bis-EMA	silanized dental glass (30–50% wt.)	Formlabs Inc., Somerville, MA, USA	A2	600163	n.a.

BisGMA = bisphenol A-glycidyl methacrylate, UDMA = Urethane dimethacrylate, BisEMA = bisphenol A ethoxylated dimethacrylate, TEGDMA = triethyleneglycol dimethacrylate, BisMEPP = bisphenol-A-ethoxylate dimethacrylate, LT = Low Translucency, HT = High Translucency, MT = Medium Translucency, OP = Opaque. n.a.= not available.

**Table 2 polymers-15-01443-t002:** Contrast Ratio (CR) of the materials tested. Materials are listed from the highest CR (opaquer) to the lowest CR (more translucent).

Product	Translucency	CR	Stat Sig
KAT	OP	84.4 ± 0.6	a
SB	LT	77.0 ± 0.8	b
BC	LT	74.1 ± 0.4	c
LU	LT	72.4 ± 0.7	d
PCR	n.d.	71.06 ± 0.4	de
KAT	LT	69.7 ± 1.6	ef
GB	LT	69.5 ± 0.7	fg
TEC	MT	66.9 ± 0.6	h
LU	HT	66.61 ± 0.3	hi
BC	HT	66.1 ± 1.2	hil
GB	HT	65.42 ± 0.7	l
CS	LT	64.8 ± 0.7	lm
SB	HT	64.8 ± 1.3	m
TEC	HT	60.0 ± 0.8	n
CS	HT	59.0 ± 1.1	n

LU = Lava Ultimate; BC = Brilliant Crios; CS = Cerasmart; KAT = Katana Avencia Block; SB = Shofu HC Block; GB = Grandio Blocs; TEC = Tetric CAD; PCR = Permanent Crown Resin; LT = Low Translucency; HT = High Translucency; MT = Medium Translucency; OP = Opaque. Different letters in Stat Sig column show significant differences among the investigate materials.

**Table 3 polymers-15-01443-t003:** Translucency Parameter (TP) of the materials tested. Materials are listed from the highest TP (more translucent) to the lowest TP (opaquer).

Product	Translucency	TP	Stat. Sig.
CS	HT	15.75 ± 0.53	a
TEC	HT	15.13 ± 0.27	b
GB	HT	14.83 ± 0.26	bc
KAT	LT	14.24 ± 0.48	cd
CS	LT	13.86 ± 0.20	de
SB	HT	13.56 ± 0.54	e
GB	LT	13.15 ± 0.28	ef
TEC	MT	12.99 ± 0.39	fg
LU	HT	12.93 ± 0.32	fg
BC	HT	12.84 ± 0.47	fg
PCR	n.d.	12.52 ± 0.19	gh
LU	LT	12.00 ± 0.29	h
BC	LT	11.00 ± 0.11	i
SB	LT	10.08 ± 0.38	l
KAT	OP	8.96 ± 0.23	m

LU = Lava Ultimate; BC = Brilliant Crios; CS = Cerasmart; KAT = Katana Avencia Block; SB = Shofu HC Block; GB = Grandio Blocs; TEC = Tetric CAD; PCR = Permanent Crown Resin; LT = Low Translucency; HT = High Translucency; MT = Medium Translucency; OP = Opaque. Different letters in Stat Sig column show significant differences among the investigate materials.

**Table 4 polymers-15-01443-t004:** Translucency Parameter 00 (TP_00_) of the materials tested. Materials are listed from the highest TP_00_ (more translucent) to the lowest TP_00_ (opaquer).

Product	Translucency	TP_00_	
CS	HT	12.47 ± 0.42	a
TEC	HT	12.41 ± 0.26	ab
GB	HT	11.76 ± 0.18	bc
KAT	LT	10.83 ± 0.40	d
CS	LT	10.66 ± 0.19	de
SB	HT	10.63 ± 0.41	de
LU	HT	10.37 ± 0.27	ef
BC	HT	10.15 ± 0.36	f
GB	LT	10.13 ± 0.23	f
TEC	MT	10.04 ± 0.22	f
PCR	n.d.	9.59 ± 0.13	g
LU	LT	9.06 ± 0.21	h
BC	LT	8.10 ± 0.09	i
SB	LT	7.49 ± 0.30	l
KAT	OP	6.31 ± 0.15	m

LU = Lava Ultimate; BC = Brilliant Crios; CS = Cerasmart; KAT = Katana Avencia Block; SB = Shofu HC Block; GB = Grandio Blocs; TEC = Tetric CAD; PCR = Permanent Crown Resin; LT = Low Translucency; HT = High Translucency; MT = Medium Translucency; OP = Opaque. Different letters in Stat Sig column show significant differences among the investigate materials.

## Data Availability

The data presented in this study are available on request from the corresponding author. The data are not publicly available due to the university’s policy on access.

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
