# Peer review of "Translucency of CAD/CAM and 3D Printable Composite Materials for Permanent Dental Restorations"

_polymers, 2023, doi:10.3390/polym15061443_

Round 1

Reviewer 1 Report

Dear Authors,

thank you for this interesting paper. Here are some suggestions how to improve it.

1. the abstract should be reevaluated - you added to much data with abbreviations, which are not clear for the reader. Only the most important information should be included in the abstract.

2. In the materials and methods I cannot see which shades of the materials the researchers used. 

3. In Materials and Methods there should be the ISO standard according to which the probes were prepared.

4. In materials and methods - please, draw a graph on how the specimens were prepared, especially 3D printed.

5. Explain all the abbreviations used in table 2 in the table title

6. In the discussion, I would add the information on the fractal dimension analysis as for future widening of the presented study (see works by Jurczyszyn, Kozakiewicz).

7. Also, refer to the color stability of the CAD-CAM, eg. 

Kul E, Abdulrahim R, Bayındır F, Matori KA, Gül P. Evaluation of the color stability of temporary materials produced with CAD/CAM. Dent Med Probl. 2021;58(2):187–191. doi:10.17219/dmp/126745

8. You should add the possible widenings of the presented study as for future planning, including the compression, tensility, eg. 

Paradowska-Stolarz A, Malysa A, Mikulewicz M. Comparison of the Compression and Tensile Modulus of Two Chosen Resins Used in Dentistry for 3D Printing. Materials (Basel). 2022 Dec 15;15(24):8956. doi: 10.3390/ma15248956

9. Please, form the conclusions as a separate chapter

I think the article is really interesting and contains many novelties. After the corrections, the article deserves publishing.

Author Response

Dear Authors,

thank you for this interesting paper. Here are some suggestions how to improve it.

  1. the abstract should be reevaluated - you added to much data with abbreviations, which are not clear for the reader. Only the most important information should be included in the abstract.

On the basis of the request of the reviewer, the abstract was partially rewritten to simplify it, trying to keep the most important informations.

  1. In the materials and methods I cannot see which shades of the materials the researchers used. 

Shade has been added to the materials’ table.

  1. In Materials and Methods there should be the ISO standard according to which the probes were prepared.

Unlike other testing methods, unfortunately, there is no ISO standard on how to prepare a specimen for evaluating translucency. In the literature, specimens vary in shape and thickness. The shape and thickness were selected based on the majority of the studies investigating translucency.

  1. In materials and methods - please, draw a graph on how the specimens were prepared, especially 3D printed.

The authors have added 3 pictures to describe the manufacturing process of the CAD/CAM and the 3D printed specimens. 3D printed specimens were created with a 3D software and then printed.

  1. Explain all the abbreviations used in table 2 in the table title

Legends explaining abbreviations have been added to footers of table 2,3 and 4.

  1. In the discussion, I would add the information on the fractal dimension analysis as for future widening of the presented study (see works by Jurczyszyn, Kozakiewicz).

Reference to Grzebieluch W, Kowalewski P, Grygier D, Rutkowska-Gorczyca M, Kozakiewicz M, Jurczyszyn K. Printable and Machinable Dental Restorative Composites for CAD/CAM Application-Comparison of Mechanical Properties, Fractographic, Texture and Fractal Dimension Analysis. Materials (Basel). 2021;14(17):4919. Published 2021 Aug 29. doi:10.3390/ma14174919 has been added outlining the need in future studies to include microhardness, fractographic, microstructural, texture and fractal dimension analyses.

  1. Also, refer to the color stability of the CAD-CAM, eg. 

Kul E, Abdulrahim R, Bayındır F, Matori KA, Gül P. Evaluation of the color stability of temporary materials produced with CAD/CAM. Dent Med Probl. 2021;58(2):187–191. doi:10.17219/dmp/126745

The reference to color stability of the CAD-CAM and Kul's reference has been added to the discussion

  1. You should add the possible widenings of the presented study as for future planning, including the compression, tensility, eg. 

Paradowska-Stolarz A, Malysa A, Mikulewicz M. Comparison of the Compression and Tensile Modulus of Two Chosen Resins Used in Dentistry for 3D Printing. Materials (Basel). 2022 Dec 15;15(24):8956. doi: 10.3390/ma15248956

A sentence including possible future correlations of the findings of the present study with compression and tensile modulus has been added together with the suggested reference.

  1. Please, form the conclusions as a separate chapter

Conclusions have been added as a separate chapter (5).

I think the article is really interesting and contains many novelties. After the corrections, the article deserves publishing.

Thanks a lot.

Reviewer 2 Report

Although the subject of the study is interesting for the readers, I believe that the following corrections should be made.

Abstract:

1.     The purpose sentence between lines 15-17 should be rewritten by expressing it more clearly. It is not understood.

2.     Findings sentence on lines 26-31 should be simpler and rewritten with textual expressions rather than numerical magnitude expressions.

3.     The conclusion sentence between lines 31-34 should be rewritten according to the findings.

Introduction:

1.     The "purpose of the study" and the following "null hypothesis of the study" sentences between lines 79-83 should be rewritten with a clearer expression. It is not understood.

Materials and methods:

1.     In Table 1, organic matrix should be used instead of matrix, and inorganic filler should be used instead of filler.

2.     Explanations of abbreviations such as BisGMA, UDMA in the Matrix column should be added to the end of the Table 1.

3.     Percentile expressions in Table 1 should always be written in the same format and should be specified for all materials.

4.     Lines 123-124 refer to the calibration of the color measurement. How many measurements was this calibration repeated? Please specify clearly.

5.     The mathematical square root expression of the TP and TP00 formulations must be written the same in both.

6.     Parametric factors in lines 153.-154 were determined as (1,1,1). In some dental studies, this is also referred to as (1,1,2). Explain the reason for choosing these expressions (1,1,1). Please add a reference about it.

Results:

1.     This section should be completely rewritten. It is not understood.

2.     The order of magnitudes between lines 170 and 171 should be rewritten in a simpler and text format.

3.     Combining Tables 2, 3, and 4 should be considered. The p value should be mentioned in these tables.

Discussion:

1.     There is no conclusion sentence. A conclusion should be added after the "Limitations of the Study" section.

Author Response

Although the subject of the study is interesting for the readers, I believe that the following corrections should be made.

Abstract:

  1. The purpose sentence between lines 15-17 should be rewritten by expressing it more clearly. It is not understood.

The sentence has been rewritten in a more understandable way.

  1. Findings sentence on lines 26-31 should be simpler and rewritten with textual expressions rather than numerical magnitude expressions.

Lines 26-31 has been rewritten in a simpler way. The textual explanation has been preferred to magnitude expressions.

  1. The conclusion sentence between lines 31-34 should be rewritten according to the findings.

Conclusions have been rewritten according to the findings.

Introduction:

  1. The "purpose of the study" and the following "null hypothesis of the study" sentences between lines 79-83 should be rewritten with a clearer expression. It is not understood.

The sentence has been rewritten. Now it is more understandable.

Materials and methods:

  1. In Table 1, organic matrix should be used instead of matrix, and inorganic filler should be used instead of filler.

Organic and Inorganic word have been added to Table 1.

  1. Explanations of abbreviations such as BisGMA, UDMA in the Matrix column should be added to the end of the Table 1.

Explanation of the monomers and other abbreviations have been added to Table1 footer.

  1. Percentile expressions in Table 1 should always be written in the same format and should be specified for all materials.

Percentile expressions have been modified. They are in the same format. Missing material has been added.

  1. Lines 123-124 refer to the calibration of the color measurement. How many measurements was this calibration repeated? Please specify clearly.

Lines 123-4 do not refer to calibration. They refer to the readings that are recorded to obtain color coordinates to calculate translucency values. The readings are performed on standardized black and white cards. The word calibrated is confusing and has been updated with "standard".

  1. The mathematical square root expression of the TP and TP00 formulations must be written the same in both.

The formulas have been uniformed.

  1. Parametric factors in lines 153.-154 were determined as (1,1,1). In some dental studies, this is also referred to as (1,1,2). Explain the reason for choosing these expressions (1,1,1). Please add a reference about it.

The parametric factor ratio was introduced as a way to control changes in the magnitude of tolerance judgments and as a way to adjust for scaling of clinical acceptability rather than perceptibility. Several authors (a, b) assumed that texture only affects lightness tolerances but not chroma or hue tolerances, and therefore the value KL = 2 was proposed.

a) Steen D, Dupont D. Defining a practical method of ascertaining textile color acceptability. Color Research and Application 2002;27:391–8.

b) Choo S, Kim Y. Effect of color on fashion fabric image. Color Research and Application 2003;28:221–6.

Nevertheless, more recent studies reported the use of 1,1,1 parametric factors as reliable. A reference has been added (Ghinea R, Pérez MM, Herrera LJ, Rivas MJ, Yebra A, Paravina RD. Color difference thresholds in dental ceramics. J Dent. 2010;38 Suppl 2:e57-e64. doi:10.1016/j.jdent.2010.07.008)

Results:

  1. This section should be completely rewritten. It is not understood.

The part of the results has been rewritten in a simpler format 

  1. The order of magnitudes between lines 170 and 171 should be rewritten in a simpler and text format.

The part of the results has been rewritten in a simpler format and a reference to the relative tables has been added.

  1. Combining Tables 2, 3, and 4 should be considered. The p value should be mentioned in these tables.

The authors initially considered to use on single table. Since the results for CR, TP and TP00 formulas are slightly different, we opted for three separate tables, each one ordered based on translucency levels. The fact to have tables ordered allowed to have three separate and more clear graphs.

Discussion:

  1. There is no conclusion sentence. A conclusion should be added after the "Limitations of the Study" section.

Conclusions have been added as a separate chapter (5).

Round 2

Reviewer 1 Report

Thank you for all the corrections improving your paper. The only thing I would suggest is adding the information on ISO standards - that they are lacking. The fact that you compared so many specimens of different materials is impressing, especially that as you responded in my review there are no unified standards. Please, add that information to the introduction (this highlightens novelity of your paper), as well as discussion. After that short correction, the paper should definately be published! Thank you so much!

Author Response

The ISO standard 7491:2000  was added in the introduction and in the discussion.

Thanks for the time spent in reviewing our article.

Reviewer 2 Report

Corrections are sufficient. It is appropriate to publish the manuscript in this way.

Author Response

Thanks for the time spent in reviewing our article.